# Magnetic Bifunctional Ru-Enzyme Catalyst Allows for Sustainable Conversion of Cellulose Derivative to D-Sorbitol

**DOI:** 10.3390/nano15100740

**Published:** 2025-05-15

**Authors:** Boris B. Tikhonov, Daniil R. Lisichkin, Alexandrina M. Sulman, Alexander I. Sidorov, Alexey V. Bykov, Yury V. Lugovoy, Alexey Y. Karpenkov, Lyudmila M. Bronstein, Valentina G. Matveeva

**Affiliations:** 1Department of Biotechnology, Chemistry and Standardization, Tver State Technical University, 22 A. Nikitina Str., 170026 Tver, Russia; tiboris@yandex.ru (B.B.T.); danok9900@gmail.com (D.R.L.); alexsulman@mail.ru (A.M.S.); sidorov_science@mail.ru (A.I.S.); bykovav@yandex.ru (A.V.B.); pn-just@yandex.ru (Y.V.L.); 2Department of Condensed Matter Physics, Tver State University, Zhelyabova St. 33, 170100 Tver, Russia; karpenkov_alex@mail.ru; 3Department of Chemistry, Indiana University, 800 E. Kirkwood Av., Bloomington, IN 47405, USA

**Keywords:** nanobiocatalyst, magnetite, nanoparticle, Ru, cellulase, cascade process

## Abstract

Here, we report the development of a novel bifunctional nanobiocatalyst for a one-pot cascade transformation of carboxymethyl cellulose (CMC) to D-sorbitol. The nanobiocatalyst is based on magnetic nanoparticle aggregates (MNAs) functionalized with chitosan (CS) cross-linked by tripolyphosphate (TPP). It contains two types of catalytic sites: cellulase (Cel, 5 wt.%) and Ru (3 wt.%) nanoparticles (NPs) of 0.7 nm in diameter. To optimize the nanobiocatalyst structure and composition, we first synthesized the biocatalyst, MNA-CSP-Cel (CSP stands for the CS layer cross-linked by TPP), as well as the nanocatalyst, MNA-CSP-Ru, and studied them in the one-step reactions of hydrolysis and hydrogenation, respectively. The data obtained allowed us to optimize the composition and properties of the bifunctional nanobiocatalyst, MNA-CSP-Ru-Cel, and to choose the best reaction conditions for the cascade process. MNA-CSP-Ru-Cel was characterized using transmission electron microscopy (TEM), high-resolution TEM, energy-dispersive spectroscopy, X-ray diffraction, X-ray photoelectron spectroscopy, and porosity measurements. The knowledge obtained enabled us to perform a cascade transformation of CMC to D-sorbitol with a yield of 83.2% for 10 h at 70 °C and a hydrogen pressure of 4 MPa. The yield demonstrated in this work is much higher than that reported to date for the same cascade process.

## 1. Introduction

Biomass processing to obtain biofuels or value-added chemicals has become a popular avenue to replace non-renewable fossil fuels, considering both socio-economic and environmental factors. On the other hand, biomass processing is a complex process, including multi-step reactions; therefore, sophisticated multifunctional catalysts are needed for a successful reaction outcome [1,2,3]. Among the numerous stages of biomass processing, the transformation of lignocellulosic biomass to sugars is the most crucial because these sugars can be further fermented into biofuels using enzymes via environmentally friendly processes [4,5,6,7,8]. However, native enzymes, including cellulases, are known for their low thermal stability, short shelf-life, and other shortcomings, including being impossible to reuse. To mitigate these shortcomings, enzymes can be immobilized on various supports [9,10]. For different processes, different supports are chosen, depending on the enzyme type and desired enzyme–support interactions, allowing one to maintain the enzyme’s secondary structure [11,12,13]. Native cellulases degrading cellulose include such enzymes as exoglucanases (EC 3.2.1.91), endoglucanases (EC 3.2.1.4), and β-glucosidases (EC 3.2.1.21), which decompose certain cellulose chemical bonds [14,15,16]. In this work, we use a single term—cellulase (Cel)—which accounts for the above enzyme complex.

Nanostructured supports of different types (nanoparticles (NPs), nanoporous supports, stimuli-responsive nanostructured supports, etc.) have received considerable attention due to improved diffusion, enhancing the mass transfer of reagents and intermediates [17,18,19,20,21,22]. Furthermore, nanostructured supports can enhance enzyme loading due to high surface areas, thus providing higher activity of such nanobiocatalysts [23,24]. Enzymes can be attached to the nanostructured support via direct conjugation, covalent attachment, electrostatic bonding, or specific coupling [15,25,26]. Among nanostructured supports, magnetic NPs (MNPs) are especially attractive for immobilization of enzymes (including celullase) because they can be magnetically recovered both during the nanobiocatalyst syntheses and in repeated use. Functionalization of MNPs allows one to enhance compatibility with enzymes as well as nanobiocatalyst stability [16,27,28,29,30,31]. Various functional groups can be utilized on the surface of MNPs, but in the majority of cases, authors describe amino groups introduced via silane/aminosilane [16,30], amino-functionalized MOFs [27], zeolites modified with amino groups [32], or amino group-containing polymers (polyethyleneimine, PEI [28]). A functionalization with carboxyl groups was also reported via meso-2,3-dimercaptosuccinic acid followed by the attachment of a melamine–glutaraldehyde dendrimer [31]. Besides glutaraldehyde, which is a common cross-linking agent for linkage of enzymes [16,30], carbodiimide has also been employed [27].

Polymers containing amino groups are excellent for modification of the MNP surface due to having multiple functional groups. However, considering that PEI precursors are quite toxic [33,34], chitosan (CS), a natural polysaccharide based on biomass-derived chitin, is the better choice for an amino group-containing polymer [35,36,37,38]. CS has become popular because it is readily available, non-toxic, hydrophilic, biocompatible, inexpensive, and easily modified with enzymes by binding to non-essential enzyme groups to preserve enzyme conformation [25]. It is noteworthy that CS-functionalized MNPs have been utilized in Cel immobilization to produce bio-alcohols [39,40,41]. In ref. [39], the authors reported higher activity of such immobilized enzymes than that of native Cel, including activity improvement in wide temperature and pH ranges.

A combination of enzymes and transition metal catalytic sites to develop multifunctional catalysts has been well demonstrated [42,43,44]. Such catalysts, containing both noble and earth-abundant metals as well as enzymes, were successful in both concurrent and sequential cascade reactions. The latter route is recommended when the catalysts are incompatible and cannot be combined on the same support [43], but a single multifunctional catalyst is preferred from the viewpoint of streamlining the process.

Here, we report novel magnetically recoverable bifunctional catalysts based on magnetite NP aggregates (MNAs) modified by CS and containing Ru NPs as well as immobilized Cel. The nanobiocatalysts synthesized in this work were tested in a cascade reaction of cellulose hydrolysis to sugars followed by their hydrogenation to D-sorbitol (hexitol) (Figure 1). D-sorbitol is a valuable compound, displaying moisturizing and emulsifying properties [45,46], and is a platform chemical for syntheses of a number of substances, such as glycerol, ethylene glycol, propylene glycol, lactic acid, etc. [47,48,49]. When D-sorbitol is obtained from biomass glucose, it is especially beneficial due to environmental considerations [50,51,52]. The majority of the research avenues in optimizing this process are focused on the development of efficient catalysts [53]. Among these, Ru-containing catalysts were found to be promising thanks to their high stability, activity, and selectivity in hydrogenation to D-sorbitol, replacing inefficient Ni-containing catalysts [54,55].

It is worth noting that a bifunctional catalyst containing acid and hydrogenation sites for obtaining D-sorbitol from cellobiose (cellulose model compound) was reported in refs. [52,56]. We employ a different approach, using 0.7 nm Ru NPs as well as enzymatic sites instead of acid sites, thus making this process more environmentally friendly and leading to a much higher yield of the target product. Previously, we reported functionalization of MNAs with a sub-nanometer (0.9 nm) CS layer cross-linked by tripolyphosphate (TPP) (designated CSP) for glucose oxidase immobilization [57]. In this work, a combination of Ru NPs with Cel on the same functionalized magnetic support resulted in a novel bifunctional nanobiocatalyst, whose properties were studied in the chemoenzymatic one-pot cascade transformation of carboxymethyl cellulose (CMC) to D-sorbitol. To better understand this process and to identify the best conditions, we fabricated the biocatalyst (MNA-CSP-Cel) for the CMC hydrolysis to D-glucose and the nanocatalyst (MNA-CSP-Ru) for the hydrogenation of D-glucose to D-sorbitol and studied the corresponding reactions. The knowledge acquired allowed us to develop the nanobiocatalyst (MNA-CSP-Ru-Cel), which showed excellent performance in the cascade process.

## 2. Materials and Methods

Materials, a detailed catalyst synthesis description, catalyst characterizations, catalytic experiments, and kinetics are presented in the Appendix A.

The synthesis of MNAs was carried out according to a procedure published elsewhere, which involves using FeCl_2_ and FeCl_3_ and co-precipitating them in basic conditions [58]. See the Appendix A for details.

Coating of MNAs with CS followed by further cross-linking with TPP is reported in our preceding paper [57]. The sample synthesized was labeled MNA-CSP. The typical experiment is described in the Appendix A.

The formation of Ru NPs on the surface of MNA-CSP was carried out via coordination of Ru(OH)Cl_3_ with CS amino groups on the support surface at 60 °C followed by reduction with NaBH_4_. A typical experiment of MNA-CSP-Ru synthesis is described in the Appendix A.

The immobilization of Cel to form MNA-CSP-Cel and MNA-CSP-Ru-Cel was carried out in the same way as that of glucose oxidase in our preceding paper [57]. See the Appendix A for the description of a typical experiment.

## 3. Results and Discussion

As a nanobiocatalyst support, we used MNAs (see Appendix A for the transmission electron microscopy (TEM) image) coated with a sub-nanometer CS layer (0.9 nm) cross-linked with TPP [57]. The magnetic measurements (Appendix A) show that the saturation magnetization of MNAs is 42 emu/g, which is consistent with that for other magnetite NPs [59]. The same support was employed to fabricate immobilized Cel (MNA-CSP-Cel) and Ru NPs in the CSP layer (MNA-CSP-Ru) to study single steps in the cascade process, e.g., the transformations of CMC to D-glucose and the latter to D-sorbitol, respectively, to identify the best catalyst composition and properties for these steps. After that, we carried out a consecutive deposition of Ru NPs and Cel on the same support and studied the catalytic performance of the nanobiocatalyst in the cascade transformation of CMC to D-sorbitol.

### 3.1. Cel Biocatalyst Catalytic Activity

In this work, Cel was covalently attached to CSP via bonding with N-hydroxysuccinimide (NHS) when carbodiimide (EDC) was present [57]. The amount of immobilized Cel was determined via testing the Cel activity before the attachment to MNA-CSP and the activity of the supernatant after the biocatalyst separation. We synthesized five biocatalyst samples containing Cel in amounts of 1.25, 2.5, 5.0, 7.5, and 10 wt.%. The different Cel contents allow one to achieve different Cel loadings in the catalytic reaction (0.05, 0.1, 0.2, 0.3, and 0.4 mg/mL, respectively) at the same biocatalyst loading to compare with loadings of native Cel.

The activity of native Cel and MNA-CSP-Cel was estimated by the quantity of glucose obtained in hydrolysis of CMC to determine optimal values of three important parameters: the substrate concentration (in the range of 1.91 to 11.45 mM), temperature (from 30 to 80 °C), and pH (3–8). We used a CMC assay method (see details in the Appendix A) to determine the above parameters.

For the enzyme concentration variation, a temperature of 50 °C and a pH of 5 (CMC concentration of 0.25% (*w*/*v*) in 25 mL of the sodium citrate buffer) were chosen based on the maximum activity of native Cel [41,60]. Figure 1 shows that upon the increase in the Cel concentration to 0.2 mg/mL (MNA-CSP-Cel with 5 wt. Cel), its activity increases and then remains unchanged at further Cel concentration increases. Thus, the optimal enzyme concentration of 0.2 mg/mL, i.e., the biocatalyst sample containing 5 wt.% Cel, was chosen for further studies. Here and below, it is referred to as MNA-CSP-Cel for simplicity. It is noteworthy that the activity of MNA-CSP-Cel is ~8% lower than that of native Cel, which is consistent with a typical decrease in the activity of immobilized enzymes [41].

We carried out a thermal gravimetric analysis (TGA) of MNA-CSP-Cel, MNA-CSP, and native Cel (Appendix A). According to the TGA data, the difference between the weight loss of MNA-CSP-Cel and MNA-CSP at 600 °C is 3.32%, which can be assigned to the Cel weight loss. However, considering that Cel preserves ~50% of its weight at 600 °C, the Cel weight loss could be 6.64%. Another concern is that the Cel decomposition can be different when it is immobilized on the MNA-CSP surface compared to free Cel; thus, no accurate value can be obtained from TGA. The correct assumption would be that the Cel content in this sample is between 3.32 and 6.64 wt.%, which is consistent with the loading amount.

Appendix A shows that Cel activity increases with the temperature for both native and immobilized enzymes and then decreases upon further increase. For the native enzyme, the highest activity was observed at 50 °C, while for MNA-CSP-Cel, the highest activity was achieved at 60 °C. Above the indicated temperatures, the activity dropped, which could be assigned to both Cel denaturation and decomposition [61]. It is worth noting, however, that though for native Cel, the activity dropped precipitously above 50 °C, for MNA-CSP-Cel, the activity was above 92% of its highest value at 70 °C, demonstrating clear stabilization of Cel upon immobilization—consistent with other reports [41,62].

The pH influence study (Appendix A) showed that for native Cel, the highest activity is achieved at pH 5, while for MNA-CSP-Cel, it is accomplished at pH 6. These data suggest that the higher activity of immobilized enzymes at a higher pH could be due to better access to Cel active sites on the MNA-CSP surface [60], indicating changes in the environment of Cel active sites upon immobilization (consistent with the previous report [63]). A further increase in pH leads to excessive basicity, destroying the Cel active sites [63,64].

To better understand hydrolysis of CMC with immobilized Cel, we carried out kinetic studies. The kinetic constants (the apparent binding constant, Km, and maximal reaction velocity, Vmax) for native Cel and MNA-CSP-Cel were determined at varying CMC concentrations (from 1.91 to 11.45 mM) at 50 °C and a pH of 5. To calculate Km and Vmax, we used Lineweaver–Burk plots (Appendix A). The Km and Vmax values for native Cel and MNA-CSP-Cel are displayed in Table 1.

The data show that the affinity of Cel to CMC reflected by the Km value significantly increases after immobilization, which is supported by the literature data [65]. On the other hand, the activity of Cel decreases after immobilization (decrease in Vmax), consistent with other reports [41,60].

The storage stability of native Cel and MNA-CSP-Cel at 4 °C was measured by estimating the relative activity after 49 days (with a 7-day interval) (Figure 2). It was demonstrated that MNA-CSP-Cel preserved 79% of its initial activity after 49 days, while native Cel retained only 63%. This is attributed to the improvement in the conformational stability of Cel after immobilization [66]. On the other hand, much lower initial activity retention (64%) was reported for Cel immobilized on CS-coated magnetic graphene (after 45 days) [60], suggesting that the conformational stability upon immobilization could be dependent on the surface curvature or some other factors.

### 3.2. Ru-Containing Nanocatalyst Performance in Hydrogenation

In this work, MNA-CSP-Ru samples with different Ru contents (Table 2) were synthesized and studied in the hydrogenation of D-glucose to D-sorbitol.

Table 2 shows that D-glucose hydrogenation to D-sorbitol occurs with very high selectivity of 99.3% for the samples with 1, 2, and 3 wt.% of Ru. However, at Ru contents of 1 and 2 wt.%, the conversion is too low. The full conversion (99.9%) is achieved for the sample containing 3 wt.% of Ru. The further increase in the Ru content to 5 wt.% results in a decrease in selectivity, probably due to isomerization of D-sorbitol to D-mannitol. Based on the above data, the MNA-CSP-Ru catalyst with 3 wt.% of Ru was chosen for characterization and further studies. Here and below, we refer to this nanocatalyst as MNA-CSP-Ru.

The comparison of the performance of various Ru catalysts in hydrogenation of D-glucose to D-sorbitol is presented in Table 3. To the best of our knowledge, MNA-CSP-Ru allows for the best combination of activity and selectivity among the catalysts.

### 3.3. Bifunctional Catalyst for a One-Pot Cascade Transformation

#### 3.3.1. Nanobiocatalyst Synthesis

As is demonstrated above, the Cel-containing biocatalyst and the Ru-containing nanocatalyst, both based on MNA-CSP, lead to successful outcomes of their respective reactions: enzymatic hydrolysis of CMC to D-glucose and hydrogenation of D-glucose to D-sorbitol. The main goal of this study is to develop a bifunctional catalyst with both Ru and Cel active sites to carry out the cascade process in the same reactor. To accomplish this, we formed Ru NPs on the MNA-CSP surface, followed by Cel immobilization (see the Appendix A for details). Figure 2 shows the sequence of the steps in the bifunctional nanobiocatalyst synthesis. It was found that the presence of Ru NPs does not influence the amount of immobilized Cel or the dependencies of the activity on temperature and pH for MNA-CSP-Ru-Cel: they were the same as those obtained for MNA-CSP-Cel (see Appendix A).

#### 3.3.2. Characterization of MNA-CSP-Ru-Cel

TEM and high-resolution TEM (HRTEM) images of MNA-CSP-Ru-Cel are presented in Figure 3. TEM shows MNAs with overlapping iron oxide NPs, while Ru NPs are not visible. The HRTEM image clearly shows small darker spots on the MNA background which could be assigned to Ru NPs. The mean diameter of Ru NPs is 0.7 nm. Hysteresis loops (Appendix A) demonstrate that the saturation magnetization of MNA-CSP-Ru-Cel (39.0 emu/g) is only 7% lower than that of MNAs (42.0 emu/g), revealing that the catalysts can be easily magnetically separated during synthesis and after the catalytic reaction.

Scanning TEM (STEM) energy-dispersive spectroscopy (EDS) maps are presented in Figure 4. The maps of individual elements as well as the superposition of the Fe, P, and Ru maps show that all elements are evenly spread on MNAs.

The EDS spectrum presented in Appendix A demonstrates the presence of the above elements along with carbon in the MNA-CSP-Ru-Cel sample. The XPS survey spectrum of MNA-CSP-Ru-Cel (Figure 5a) shows that the nanobiocatalyst surface contains the same elements as those identified by EDS.

The contents of each element in MNA-CSP-Ru-Cel are given in Appendix A. XPS shows much higher Ru content on the surface (5.49 wt.%) than the elemental analysis (3.11 wt.%; see the Appendix A), which is a bulk method. This means that the surface is enriched with Ru—consistent with the formation of Ru NPs in the CSP layer. The deconvolution (Appendix A) of the high-resolution Ru 3d XPS spectrum of MNA-CSP-Ru-Cel exhibits only one type of the oxidation state, Ru^4+^ (from RuO_2_ species), on the nanobiocatalyst surface, demonstrating that a complete surface oxidation occurs despite the Ru NP formation being carried out with a strong reducing agent—NaBH_4_. Formation of RuO_2_ on the Ru NP surface was previously reported [71]. The X-ray diffraction (XRD) pattern of MNA-CSP-Ru-Cel (Appendix A) is characteristic of the spinel structure and contains no reflections assigned to Ru metal or RuO_2_ crystals which could be attributed to the small size of Ru NPs or their amorphous character.

Porosity measurements were carried out using liquid nitrogen adsorption for MNAs, MNA-CSP, and MNA-CSP-Ru-Cel. The adsorption–desorption isotherms of these materials (Appendix A) are typical for mesoporous solids (type IV isotherms) [72]. The data presented in Appendix A show that the CSP layer deposition results in a significant decrease in the BET (Brunauer–Emmett–Teller) surface area from 220 to 150 m^2^/g (by ~32%), while the pore volume decreases by only about 7%. Appendix A demonstrates that the 3–6 nm pores of MNAs are significantly blocked while new, larger pores are formed, most likely due to spaces between MNA particles connected by the CSP layer. The formation of Ru NPs in the CSP layer as well as Cel immobilization results in an insignificant decrease in the BET surface area (~13%), but a noticeable decrease in the pore volume (~34%). This would be consistent with the clogging of some pores either due to side products of the NaBH_4_ reduction or due to Cel attached in pore junctions.

STEM EDS maps and the HRTEM image (Appendix A) were also obtained for MNA-CSP-Ru for assessment of the Ru NP size and the distribution of all the species before and after Cel attachment. These data show that MNA-CSP-Ru is unaffected by the attachment of Cel.

#### 3.3.3. MNA-CSP-Ru-Cel Performance in the Cascade Process

Figure 3 shows the one-pot chemoenzymatic cascade transformation of CMC to D-sorbitol with MNA-CSP-Ru-Cel. The major problem of the one-pot cascade reaction is that the optimal reaction conditions of the enzymatic reaction and hydrogenation are very different. The enzymatic hydrolysis of CMC to D-glucose occurs efficiently at 50–70 °C and pH 5–7, while hydrogenation of D-glucose to D-sorbitol with high activity and selectivity takes place at 100 °C, a hydrogen pressure of 4 MPa, and pH 7. It is worth noting that such dissimilarity in reaction conditions for different steps of cascade reactions leads researchers to use two-pot processes instead of the one-pot reaction, as was reported for transformations of corncob to furfuryl alcohol [73], a holocellulose fraction to hydroxymethylfurfural and furfural [74], and a bamboo shoot shell to furfuryl alcohol [75].

For the one-pot chemoenzymatic cascade transformation of CMC to D-sorbitol with MNA-CSP-Ru-Cel, we chose to carry out the reaction at 70 °C and pH 7—the highest values allowing one to preserve Cel conformational integrity. Table 4 shows the composition of the reaction mixture after 5.0, 7.5, and 10.0 h. The full CMC conversion is achieved after 7.5 h, but at this reaction time the D-glucose conversion is only 70.9%, with a selectivity to D-sorbitol of 98.8%. The chromatogram can be seen in the Appendix A (Appendix A). Clearly, at 70 °C, hydrogenation of D-glucose to D-sorbitol proceeds slowly, which is consistent with the literature data [76]. After 10 h, the full conversion of D-glucose is obtained; however, the selectivity to D-sorbitol drops to 83.2%, most likely due to isomerization of D-sorbitol to D-mannitol when the reaction time increases [52,76]. The carbon mass balance confirms that no other products were formed (see the Appendix A).

The blank runs with MNA-CSP and MNA-CSP-Cel showed that these compounds are inactive in the cascade reaction. It is worth noting that MNA-CSP-Ru-Cel shows similar results (to those for MNA-CSP-Ru) in the hydrogenation of D-glucose to D-sorbitol, i.e., D-glucose conversion of 99.9% and selectivity to D-sorbitol of 99.3%.

Recycling of the MNA-CSP-Ru-Cel catalyst was studied in five consecutive cycles under the conditions indicated above. After each cycle, the nanobiocatalyst was separated from the reaction mixture with a rare earth magnet and washed with a buffer solution (pH 7). Figure 6 shows that the nanobiocatalyst is comparatively stable, exhibiting decreases in activity by only 8.8% (from 100% to 91.2%) and in selectivity by only 8.4% (from 83.2% to 74.8%) after five catalytic cycles.

In recycling experiments, the reaction solutions obtained after the catalyst magnetic separation were analyzed by atomic absorption spectroscopy (AAS) to determine whether leaching of Ru occurs or not. After the first catalytic cycle, only 0.5% of the total Ru amount (14 mcg) was found in the solution, while after the subsequent cycles, only traces of Ru were seen (see the Appendix A). The negligible leaching after the first catalytic reaction can be ascribed to the loss of loosely bound Ru, while in further experiments, the catalyst was exceptionally stable. The survey XPS spectrum of the catalyst after the first catalytic reaction (Appendix A) shows a slight redistribution of the element amounts, particularly an enrichment with carbon, which is consistent with adsorption of reactants or products during the reaction.

It is noteworthy that the result obtained here, i.e., a D-sorbitol yield of 83.2% (at 70 °C and a hydrogen pressure of 4 MPa for 10 h), significantly exceeds the published data for a bifunctional Ru/CCD-SO_3_H catalyst (sulfonic acid-functionalized carbonized cassava dregs containing Ru), for which the D-sorbitol yield in the same cascade reaction was 63.8% at 180 °C (other conditions were the same) [52]. In our opinion, the higher activity of MNA-CSP-Ru-Cel is due to the small size of Ru NPs, allowing for close contact with the CSP environment, leading to a better reaction outcome in much milder conditions.

## 4. Conclusions

In this work, we developed a novel bifunctional nanobiocatalyst based on a functionalized magnetic support, MNA-CSP, and containing Ru NPs and immobilized Cel (MNA-CSP-Ru-Cel). This catalyst was designed for a one-pot chemoenzymatic cascade transformation of CMC to D-sorbitol which proceeds via two steps: enzymatic hydrolysis of CMC to D-glucose with immobilized Cel and hydrogenation of D-glucose to D-sorbitol with the Ru catalyst. To optimize each step as well as the nanobiocatalyst composition, we synthesized a biocatalyst, MNA-CSP-Cel, and nanocatalyst, MNA-CSP-Ru, and studied them in the respective single reaction steps. We determined the optimal Cel loading in MNA-CSP-Cel and the Ru content in MNA-CSP-Ru as well as the best reaction conditions for each catalytic step. The data obtained allowed us to fabricate MNA-CSP-Ru-Cel with the best Cel and Ru loadings and to choose suitable reaction conditions for the cascade process. This research resulted in a successful cascade reaction of CMC to D-sorbitol, with a D-sorbitol yield of 83.2% (at 70 °C, pH 7, and a hydrogen pressure of 4 MPa for 10 h), which significantly exceeds the published data for the same process. A full characterization of MNA-CSP-Ru-Cel allowed us to understand structure–property relationships. We demonstrated that 0.7 nm Ru NPs evenly distributed through the nanobiocatalyst do not prevent Cel immobilization and are quite efficient even at lower than optimal reaction temperatures. We believe this work paves the way for the development of various bifunctional chemoenzymatic catalysts for one-pot cascade transformations.

## Data Availability

Research data are available from the authors upon request.

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
