# Peer review of "Magnetic Bifunctional Ru-Enzyme Catalyst Allows for Sustainable Conversion of Cellulose Derivative to D-Sorbitol"

_nanomaterials, 2025, doi:10.3390/nano15100740_

Round 1

Reviewer 1 Report

Comments and Suggestions for Authors

Bronstein, Matveera, and collaborators synthesized and characterized the catalyst MNA-CSP-Ru-Cel, a bifunctional nanobiocatalyst designed for the one-pot cascade transformation of carboxymethyl cellulose (CMC) into D-sorbitol. The main components of the catalyst, MNA-CSP-Cel and MNA-CSP-Ru, were individually investigated for their roles in CMC hydrolysis to D-glucose and hydrogenation to D-sorbitol, respectively. Overall, the manuscript is well-written and offers valuable insights to the scientific community. However, a major revision is currently required to enhance the overall quality of the manuscript. Below are some comments aimed at improving its clarity and impact.

Abstract:

1) Please inform which characterizations were performed.

2) Provide specific amount of Ru and others significant parts (eg. Cel and etc) of the catalyst.

Abstract/Introduction:

3) The authors states: […] ultra-small Ru nanoparticles […]; […] very small Ru NPs […]; […]; […] ultra-thin CS layer […]. This information is relative. Therefore, I recommend removing it.

Results and discussion:

4) Is it necessary 2 hours? A kinect profile would be interesting.

5) Was leaching of the catalyst observed in the hydrogenation of D-glucose to D-sorbitol using MNA-CSP-Ru? Please provide an ICP OES analysis.

6) The authors states “[…] MNA-CSP-Ru-Cel: they were the same as those obtained for MNA-CSP-Cel (the data are not shown).” Please provide data.

7) The investigation of MNA-CSP-Ru-Cel in the hydrogenation of D-glucose to D-sorbitol can bring insights in the discussion. I also suggest the investigation of MNA-CS, MNA-CSP and MNA-CSP-Cel.

8) The authors states “The X-ray diffraction (XRD) pattern of MNA-CSP-Ru-Cel (the data are not shown) […]”. Please provide the data.

9) The product D-mannitol is also interesting. Can this product be obtained in high yields in longer reaction times? A kinect profile would also be interesting for discussion. Please comment on the mass balance.

10) Was leaching of the catalyst observed in the recycling experiments with MNA-CSP-Ru-Cel? Please provide an ICP OES analysis.

11) It would be interesting to characterize the catalyst after the reaction (or during catalyst recycling) to verify possible structural changes that justify the decrease in conversion and selectivity for the desired product.

12) Please compare whether the one-pot procedure is more efficient than the one-step process in terms of yield to the desired product.

13) Please provide TOF and TON.

Supporting information:

14) What is the yield for the “Synthesis of MNA”?

15) How much MNA is used in the procedure "Modification of MNA with chitosan and tripolyphosphate"? What is the yield? Provide characterization.

16) How much MNA-CSP is used in the procedure " Synthesis of MNA-CSP-Ru"? What is the yield? Provide characterization.

17) Check for typos

18) Provide experimental conditions for the Figure S2-S6

19) Figure S5 and S6, please use the same time scale.

20) Please provide characterization data for the products D-sorbitol and D-mannitol.

Author Response

Abstract:

Comment 1: Please inform which characterizations were performed.

Response 1: The characterization methods are now listed in the Abstract.

Comment 2: Provide specific amount of Ru and others significant parts (eg. Cel and etc) of the catalyst.

Response 2: Specific amounts of Ru (3 wt.%) and Cel (5 wt.%) have been added in the Abstract.

 Abstract/Introduction:

Comment 3: The authors states: […] ultra-small Ru nanoparticles […]; […] very small Ru NPs […]; […]; […] ultra-thin CS layer […]. This information is relative. Therefore, I recommend removing it.

Response 3: The relative information was removed and replaced by quantitative data.

Results and discussion:

Comment 4: Is it necessary 2 hours? A kinect profile would be interesting.

Response 4: Two hours are necessary for a full D-glucose conversion, while at shorter times, the lower conversion is achieved. A kinetic profile has been added to the Supplementary Materials (Figure S8).

Comment 5: Was leaching of the catalyst observed in the hydrogenation of D-glucose to D-sorbitol using MNA-CSP-Ru? Please provide an ICP OES analysis.

Response 5: ICP OES is not available to us, but we used another reliable method, the atomic absorption spectroscopy (AAS) to study the reaction solution after the catalyst separation (see https://doi.org/10.1002/aoc.590030603, https://doi.org/10.1016/j.foodchem.2005.12.033). According to the AAS, after the first catalytic cycle, the catalyst loses only 0.5% of all Ru (14 mcg), and then only traces of Ru were found in the subsequent reaction solutions. The small leaching in the first catalytic cycle can be due to desorption of weakly bound Ru during the initial reaction. After that the Ru catalyst is remarkably stable.

Comment 6: The authors states “[…] MNA-CSP-Ru-Cel: they were the same as those obtained for MNA-CSP-Cel (the data are not shown).” Please provide data.

Response 6: The requested information has been added in the revised manuscript (page 7) and in the Supplementary Materials (Figures S3-S4).

Comment 7: The investigation of MNA-CSP-Ru-Cel in the hydrogenation of D-glucose to D-sorbitol can bring insights in the discussion. I also suggest the investigation of MNA-CS, MNA-CSP and MNA-CSP-Cel.и

Response 7: It was found that MNA-CSP-Ru-Cel shows similar results in the hydrogenation of D-glucose to D-sorbitol, i.e., D-glucose conversion of 99.9% and selectivity to D-sorbitol of 99.3%. These data have been added to the revised manuscript on page 12. Also, in the Supplementary Materials we added information on blank runs for MNA-CSP and MNA-CSP-Cel in the same reaction. The sample MNA-CS is not isolated as it is immediately treated with TPP.

Comment 8: The authors states “The X-ray diffraction (XRD) pattern of MNA-CSP-Ru-Cel (the data are not shown) […]”. Please provide the data.

Response 8: The XRD pattern has been added to the Supplementary Materials (Figure S10).

Comment 9: The product D-mannitol is also interesting. Can this product be obtained in high yields in longer reaction times? A kinect profile would also be interesting for discussion. Please comment on the mass balance.

Response 9: To the best of our knowledge, the longer reaction times still produce a mixture of D-sorbitol and D-mannitol. Please note that the cascade transformation of CMC to D-sorbitol is a complex process and its kinetics is not yet studied. The kinetic profile of D-glucose conversion is presented in the Supplementary Materials (Figure S8). The carbon mass balance description has been added to the Supplementary Materials (page 10) and mentioned in the main text on page 11.

Comment 10: Was leaching of the catalyst observed in the recycling experiments with MNA-CSP-Ru-Cel? Please provide an ICP OES analysis.

Response 10: As was discussed in the response to comment 5, we used the AAS to identify leaching. After the first catalytic cycling 0.5% of all Ru was lost, and only traces of Ru were found after the subsequent catalytic cycles. This information is added to page 12 of the revised manuscript.

Comment 11: It would be interesting to characterize the catalyst after the reaction (or during catalyst recycling) to verify possible structural changes that justify the decrease in conversion and selectivity for the desired product.

Response 11: After the catalytic reaction MNA-CSP-Ru-Cel was studied by XPS. The data presented in Table S1 show insignificant redistribution of the element amounts. One can see that the increase of carbon on the catalyst surface (by 8.4 wt.%) is most likely due to deposition of the substrates and products. This discussion has been added to page 12 of the main text.

Comment 12: Please compare whether the one-pot procedure is more efficient than the one-step process in terms of yield to the desired product.

Response 12: The goal of this work is to study a one-pot procedure. Surely, the separate one-step reactions can be more efficient (as was also shown by us), because they can be carried out in optimal conditions, while the one-pot procedure always requires a compromise. But this is a valuable compromise due to economic and environmental factors.

Comment 13: Please provide TOF and TON.

Response 13: Kinetics of the cascade process is not yet studied so TOF cannot be calculated. For the D-glucose hydrogenation to D-sorbitol, TOF is 0.024 s1 (86 h-1) which exceeds by a factor of 2.5 the value of TOF obtained for Ru-ZSM-5 in ref. Selective hydrogenation of D-glucose to D-sorbitol over Ru/ZSM-5 catalysts, Chinese Journal of Catalysis, 35 (5), 2014, 733-740, https://doi.org/10.1016/S1872-2067(14)60077-2

We do not have sufficient data to perform TON calculation. Besides, this parameter is not characteristic of this reaction type and is not provided in the literature.

Supporting information:

Comment 14: What is the yield for the “Synthesis of MNA”?

Response 14: In this experiment the MNA yield was 95% from the calculated amount (see page 1 of the Supplementary Materials).

Comment 15: How much MNA is used in the procedure "Modification of MNA with chitosan and tripolyphosphate"? What is the yield? Provide characterization.

Response 15: MNA is not separated from the reaction mixture, so it is not weighed (see the procedure). The yield in this reaction is 93% (see page 1 of the Supplementary Materials). The full characterization is given in our preceding paper (Magnetic Nanoparticle Support with an Ultra-Thin Chitosan Layer Preserves the Catalytic Activity of the Immobilized Glucose Oxidase. Nanomaterials 2024, 14, 700.https://doi.org/10.3390/nano14080700) which is in Open Access.

Comment 16: How much MNA-CSP is used in the procedure " Synthesis of MNA-CSP-Ru"? What is the yield? Provide characterization.

Response 16: We used 1 g of MNA-CSP and the yield of MNA-CSP-Ru is 94% (see page 1 of the Supplementary Materials). The HRTEM image and STEM EDS maps of MNA-CSP-Ru are provided as Figures S12&S13. The discussion is added on page 10 of the revised manuscript.

Comment 17: Check for typos

Response 17: The text was checked for typos and corrected.

Comment 18: Provide experimental conditions for the Figure S2-S6

Response 18: The experimental conditions for Figures S2-S6 have been added.

Comment 19: Figure S5 and S6, please use the same time scale.

Response 19: The figures have been corrected.

Comment 20: Please provide characterization data for the products D-sorbitol and D-mannitol.

Response 20: The chromatogram is now added to the Supplementary Materials (Figure S14).

Reviewer 2 Report

Comments and Suggestions for Authors

In this manuscript, the authors demonstrate a method for the preparation of heterogenized, bifunctional, chemoenzymatic catalysts based on the use of components such as Fe-containing magnetic nanoparticles as carriers and ultra-small Ru nanoparticles in combination with the cellulase enzyme. The as- prepared and well-characterised systems appear to be very efficient catalysts for the conversion of carboxymethylcellulose (CMC) to D-sorbitol via a reaction cycle involving enzyme-catalysed hydrolysis and Ru-promoted hydrogenation steps. The entire study deserves to be published; however, I cannot step away from some weaknesses of this study which are needed to clarify before acceptance. My concerns and criticisms are below:

1, To confirm the heterogeneity of this envisioned and performed multistep catalytic transformation, the authors need to test their system with a hot filtration test. To properly accomplish this test, follow (and cite) the protocol of the following study: https://doi.org/10.1016/j.mtchem.2024.102440.

2, How to ensure that the individual components (MNP or cellulase or Ru-NP, etc.) or any physical mixture of these components cannot promote the entire multicomponent catalytic cycle with the same efficiency as the presented heterogeneous catalysts in the absence of the other components? The effectiveness of the aforementioned components and mixtures needs to be tested.

3, How was it possible to avoid the occurrence of fructose and GVL in large amounts during the multistep processes?

4, How can the heterogenized cellulase work efficiently at pH = 7 in the multistep cycle? Is there a structural, steric or morphological reason for this phenomenon? This question is really interesting considering that the native enzyme functions well at a pH of 5.

5, The actual amount of Ru and other noble metals as possible impurities should be determined e.g. by ICP-MS (OES) technique. (EDX is semi-quantitative, while XPS is only sensitive to surface composition.)

5, On the basis of the well-established terminology of multicomponent reactions, I am sure that the authors need to reconsider the use of the term “cascade”. (I suggest the use of the term “tandem”.) See: https://doi.org/10.1016/j.ccr.2004.05.012

Considering my above-detailed concerns and criticisms, I suggest the acceptance of this manuscript after a major revision.

Author Response

Comment 1: To confirm the heterogeneity of this envisioned and performed multistep catalytic transformation, the authors need to test their system with a hot filtration test. To properly accomplish this test, follow (and cite) the protocol of the following study: https://doi.org/10.1016/j.mtchem.2024.102440.

Response 1: The hot filtration test (here it is hot magnetic separation) has been performed, and the results are now added to the Supplementary Materials. Hydrogenation was carried out in optimal conditions and MNA-CSP-Ru-Cel was isolated after 1 h (at a half yield of the product) using a rare earth magnet. After that, the reaction solution was allowed to be in the reaction conditions (in the absence of the catalyst) and the product distribution was analyzed. It was found that after the catalyst removal, no product was formed, indicating the heterogeneous character of this process.  

Comment 2: How to ensure that the individual components (MNP or cellulase or Ru-NP, etc.) or any physical mixture of these components cannot promote the entire multicomponent catalytic cycle with the same efficiency as the presented heterogeneous catalysts in the absence of the other components? The effectiveness of the aforementioned components and mixtures needs to be tested.

Response 2: We added the data of blank runs with MNA-CSP and MNA-CSP-Cel to the Supplementary Materials to insure that they are inactive in this reaction. See the Supplementary Materials (page 3) and the main text (page 12).

Comment 3: How was it possible to avoid the occurrence of fructose and GVL in large amounts during the multistep processes?

 Response 3: It is noteworthy that neither fructose nor GVL were formed in the multistep process of the conversion of CMC to D-sorbitol that could be assigned to the low reaction temperature (70 °C) and pH 7.

Comment 4: How can the heterogenized cellulase work efficiently at pH = 7 in the multistep cycle? Is there a structural, steric or morphological reason for this phenomenon? This question is really interesting considering that the native enzyme functions well at a pH of 5.

Response 4: In the original manuscript we discussed that the pH influence study (Figure S4) showed that for native Cel, the highest activity (100%) is achieved at pH 5, while for MNA-CSP-Cel, it is accomplished at pH 6 (99.9%). This was attributed to better access to Cel active sites on the MNA-CSP surface so the higher pH is tolerated. This finding is consistent with the literature data presented in the manuscript. According to our data, immobilized Cel retains 96% of its activity at pH 7, utilized in the cascade process.

Comment 5: The actual amount of Ru and other noble metals as possible impurities should be determined e.g. by ICP-MS (OES) technique. (EDX is semi-quantitative, while XPS is only sensitive to surface composition.)

Response 5: The amount of Ru with high accuracy was determined by X-ray fluorescence analysis (XFA) using a Spectroscan Max spectrometer. See the Supplementary Materials (page 1).

Comment 6: On the basis of the well-established terminology of multicomponent reactions, I am sure that the authors need to reconsider the use of the term “cascade”. (I suggest the use of the term “tandem”.) See: https://doi.org/10.1016/j.ccr.2004.05.012

Response 6: We checked the suggested review article, and we do not see any proof for using the one term and not the other. We believe the terms “cascade” and “tandem” reactions are synonymous as was discussed in our recent review “Multifunctional Catalysts for Cascade Reactions in Biomass Processing” (https://doi.org/10.3390/nano14231937).

Round 2

Reviewer 1 Report

Comments and Suggestions for Authors

I really appreciate the authors' dedication to making the modifications and improving the manuscript. The manuscript can be accepted in this form.